# Exploring Possible Diagnostic Precancerous Biomarkers for Oral Submucous Fibrosis: A Narrative Review

**DOI:** 10.3390/cancers15194812

**Published:** 2023-09-30

**Authors:** Jie-Ru You, Ya-Ting Chen, Chia-Yu Hsieh, Sin-Yu Chen, Tzu-Yao Lin, Jing-Syuan Shih, Guan-Ting Chen, Sheng-Wei Feng, Tzu-Yu Peng, Chia-Yu Wu, I-Ta Lee

**Affiliations:** 1School of Dentistry, College of Oral Medicine, Taipei Medical University, Taipei 110301, Taiwan; b202111019@tmu.edu.tw (J.-R.Y.); b202111028@tmu.edu.tw (Y.-T.C.); b202111035@tmu.edu.tw (C.-Y.H.); b202111033@tmu.edu.tw (S.-Y.C.); b202111052@tmu.edu.tw (T.-Y.L.); b202111015@tmu.edu.tw (J.-S.S.); b202111067@tmu.edu.tw (G.-T.C.); shengwei@tmu.edu.tw (S.-W.F.); ypeng@tmu.edu.tw (T.-Y.P.); 2Division of Oral and Maxillofacial Surgery, Department of Dentistry, Taipei Medical University Hospital, Taipei 11031, Taiwan; 3School of Dental Technology, College of Oral Medicine, Taipei Medical University, Taipei 11031, Taiwan

**Keywords:** oral submucous fibrosis, biomarker, precancer malignant transformation, oral epithelium, microvessel density

## Abstract

**Simple Summary:**

Oral submucous fibrosis is a progressive oral disorder categorized as a potentially malignant condition. Researchers have identified specific molecular biomarkers associated with oral submucous fibrosis. The objective of our study was to conduct an up-to-date and comprehensive review of molecular biomarkers linked to oral submucous fibrosis, as investigated in studies conducted within the last five years. We organized potential molecular biomarkers associated with oral submucous fibrosis and categorized them based on cytological characteristics and sampling methodologies. To aid in the development of practical diagnostic methods for detecting oral submucous fibrosis, our study provides fellow researchers with a coherent and informative overview of these molecular biomarkers.

**Abstract:**

Oral submucous fibrosis (OSF) stands as a progressive oral ailment, designated as a potentially malignant disorder. OSF has gained widespread recognition as a significant precursor to malignant transformation. In the pursuit of dependable, straightforward, and non-invasive diagnostic measures for the early detection of oral malignant progression, research has delved into potential diagnostic biomarkers of OSF. This comprehensive review delves into current investigations that explore the correlation between various biomarkers and OSF. The molecular biomarkers of OSF are categorized based on cytology and sampling methods. Moreover, this review encompasses pertinent studies detailing how these biomarkers are acquired and processed. Within this scope, we scrutinize four potential biomarkers that hold the promise of facilitating the development of diagnostic tools for detecting early-stage OSF.

## 1. Introduction

The human oral cavity is a complex and vital part of our anatomy, encompassing various structures that facilitate essential functions such as digestion, communication, and protection. At the forefront of this intricate system is the oral mucous membrane, a moist lining that coats the oral cavity [1,2]. This pliable structure not only connects externally with the lips and skin but also internally with the pharynx, throat, and esophagus [3,4,5]. Among its multifaceted roles in humans are protection, sensation, and secretion, rendering it a crucial component of our oral health [1]. Serving as a protective barrier, the oral mucosa shields underlying tissues from the abrasive forces of biting and chewing, as well as potential viral or bacterial threats [3]. Comprising three primary layers—the epithelium, lamina propria, and submucosa—the oral mucosa’s complexity is more than skin deep. The epithelium, primarily composed of keratinocytes, forms a stratified squamous epithelium. Keratinization levels vary based on location and the functional and mechanical demands placed on specific areas [6]. Beneath the epithelium lies the connective tissue known as the lamina propria, divided into the papillary and reticular layers, both containing vital blood vessels and nerves essential for maintaining oral mucosal health and sensitivity. The deepest layer, the submucosa, consists of fibro-collagenous and elastic tissues, housing critical blood vessels, nerves, and even additional elements like adipose tissue, minor salivary glands, lymphoid tissue, and muscles depending on location. These diverse components harmoniously contribute to the oral mucosa’s overall structure and function, allowing it to endure daily mechanical stresses while remaining sensitive and responsive to various stimuli.

Oral submucous fibrosis (OSF) is a chronic condition intricately linked to malignancy [7]. It is regarded as a precancerous ailment with the ominous potential to progress into oral cancer (Figure 1). This affliction predominantly arises from the habitual consumption of betel nuts. Within the oral submucosa of affected patients, abnormal collagen accumulates at an alarming rate, primarily driven by specific constituents found in betel nuts, such as arecoline, polyphenols, crude fibers, and adipose elements [8,9]. These components not only trigger excessive collagen production within cells but also impede the cells’ capacity to effectively break down collagen. Adding to the gravity of the situation, arecoline has been implicated in genetic mutations, consequently inducing cytotoxic transformations in fibroblasts. This intricate interplay of factors underscores the progression of OSF from a benign condition to a potentially life-threatening one. The excessive collagen deposition and compromised collagen breakdown create an environment conducive to the eventual transition to oral cancer (Figure 1). As such, OSF serves as a sobering reminder of the profound impact that certain habits and substances can exert on our health, extending even to the genetic level. Efforts to curtail the advancement of OSF must, therefore, focus not only on treating its immediate manifestations but also on addressing the underlying contributors, primarily the consumption of betel nuts. Additionally, early detection and intervention are crucial in preventing the progression to oral cancer, making regular oral health check-ups imperative for individuals at risk. Moreover, raising awareness about the dangers of betel nut consumption through public health campaigns and education is essential in preventing new cases of OSF and reducing its prevalence in regions where it is prevalent. By taking a multi-pronged approach that combines medical treatment, prevention strategies, and public awareness, we can hope to combat the insidious threat posed by OSF and reduce its association with oral cancer.

The clinical manifestations of OSF include ulceration and xerostomia, as highlighted in previous studies [10]. The progression of the condition is characterized by a loss of elasticity in the oral mucosa, resulting in fibrosis and ultimately leading to difficulties in opening the mouth. This impairment significantly impacts daily life by causing difficulties in eating and speaking. Patients also experience altered taste perception, dry mouth, and heightened sensitivity to spicy or hot foods. Additionally, the limited ability to open the mouth makes it challenging to maintain proper oral hygiene, often resulting in dental issues such as cavities, toothache, and periodontal diseases. It is worth noting that these oral health issues can potentially contribute to the deterioration of overall oral well-being and might even trigger systemic diseases such as diabetes and heart diseases [11,12,13]. Given the substantial impact of OSF on individuals’ quality of life and the potential for serious health implications, the early detection of this condition is of utmost importance [14,15,16].

To this end, this article aims to comprehensively review the various biomarkers that have been utilized for the identification of OSF [3]. By conducting this review, researchers can conduct a comparative analysis of different biomarkers, evaluating their respective advantages and disadvantages. The insights gleaned from this review can then be consolidated into a report that holds the potential to inform future studies on OSF. Furthermore, as research in this field advances, it is essential to delve deeper into the understanding of these biomarkers, their potential diagnostic value, and their correlation with the progression of OSF. By shedding light on the diverse biomarkers, this article not only contributes to the scientific community’s understanding of the condition but also lays the foundation for the development of more accurate and efficient diagnostic techniques. Ultimately, early detection facilitated by robust biomarker identification could lead to timely interventions, thereby improving patient outcomes and mitigating the impact of OSF on oral and systemic health.

## 2. Biomarkers of OSF

### 2.1. Clinical Roles of Potential Biomarkers

Biomarkers, which are measurable and objective indicators, play a crucial role in assessing various aspects of biological processes [17]. In the case of OSF, they have garnered significant attention in recent years as researchers seek to identify reliable markers for this condition. OSF is a potentially debilitating oral disease characterized by fibrosis of the submucosal tissues, and while clinical symptoms and pathological examinations are currently the mainstays of diagnosis, integrating biomarkers into the diagnostic process holds the promise of more efficient and accurate assessments. Several diverse biological techniques have been employed to discover potential biomarkers for OSF. These methods encompass the examination of cytological features, analysis of promoter methylation patterns, exploration of genetic polymorphisms, evaluation of mRNA and microRNA profiles, investigation of non-coding RNAs, and the assessment of proteins and trace elements within solid tissue biopsies. Furthermore, biomarkers have even been explored in liquid samples obtained from serum and saliva, as summarized in Table 1. Incorporating biomarkers into the diagnosis of OSF offers several advantages. It not only enhances the efficiency of the diagnostic process but also conserves valuable time and resources. Beyond their diagnostic utility, the expression levels of individual or multiple biomarkers can also be used to establish a novel staging method for assessing the severity of OSF. This has the potential to greatly improve the OSF evaluation index, providing clinicians with a more comprehensive picture of the disease’s progression. Moreover, biomarkers hold promise in predicting the likelihood of OSF evolving into a malignant tumor. Identifying patients at higher risk of malignant transformation is a crucial step towards developing more targeted gene therapies and interventions, potentially preventing the onset of more severe complications. The integration of biomarkers into the diagnosis and evaluation of OSF represents a promising avenue for enhancing both diagnostic accuracy and the effectiveness of treatment strategies. This multidisciplinary approach not only aids in diagnosing OSF more effectively but also contributes to a deeper understanding of its pathogenesis. By guiding more precise therapeutic interventions, biomarkers are at the forefront of efforts to combat this challenging oral disease. As research in this field continues to evolve, the potential for improved patient outcomes and a better grasp of OSF’s underlying mechanisms becomes increasingly achievable.

### 2.2. A Potential Protein Biomarker, Ki67 

Ki67 is a non-histone nuclear and nucleolar protein encoded by the MKI67 gene. During the G1 phase of the cell cycle, Ki67 predominantly localizes to the nucleolus region surrounding the cell nucleus. It is believed that Ki67 functions somewhat like a surfactant, facilitating chromosome movement and interacting with the mitotic spindle to prevent the collapse of chromosomes into a chromatin mass after the disassembly of the nuclear membrane [39]. Ki67 serves as a marker of cell proliferation [40], and multiple experiments have confirmed its presence in the S-phase, G2-phase, and M-phase, except for the G0-phase [41]. However, there is variability in the expression of the Ki67 antigen during the G1 phase [42]. Furthermore, during the G1 and G0 phases of the cell cycle, Ki67 is continuously degraded, regardless of how cells enter G0 [43]. Consequently, the levels of Ki67 can vary significantly among individual cells during G0 and G1, depending on the duration of time each cell has spent in G0. This suggests that the presence or absence of Ki67 is not binary but rather a graded indicator [44]. Due to its role in cell proliferation, Ki67 is closely associated with cellular growth [45]. The expression of the Ki67 antigen is used to determine the proportion of cells with proliferative potential in tumors. Tumor cells with higher Ki67 expression tend to exhibit greater proliferation and local invasion potential. As a result, Ki67 is considered one of the most valuable markers when assessing the biological aggressiveness of tumors [46]. Ki67 immunohistochemistry is anticipated to have three primary applications. Firstly, it can be employed to estimate the prognosis of early-stage diseases and guide decisions regarding the necessity for adjuvant chemotherapy. Secondly, Ki67 IHC may predict the efficacy of chemotherapy and assist in treatment selection. Lastly, it can be used to monitor treatment response during or after neoadjuvant endocrine or chemotherapy, aiding in the evaluation of treatment effectiveness. These advancements in the utilization of Ki67 IHC have the potential to enable informed treatment decisions, optimize patient care, and potentially enhance overall treatment outcomes [44]. In addition to its role as a cell proliferation marker and its applications in cancer research and treatment, Ki67 has also sparked interest in the field of regenerative medicine. Researchers are exploring its potential use as a marker for assessing the proliferative capacity of stem cells and their derivatives, which could have implications for tissue engineering and regenerative therapies. Furthermore, understanding the regulation of Ki67 expression and its interactions with other cell cycle proteins continues to be a topic of active research. Elucidating the molecular mechanisms behind Ki67′s functions could lead to novel therapeutic approaches for controlling cell proliferation and potentially treating conditions where uncontrolled cell growth is a concern, such as cancer and certain autoimmune diseases. In summary, Ki67 is a multifaceted protein with a central role in cell proliferation, cancer biology, and potential applications in regenerative medicine. Its nuanced expression patterns and interactions within the cell cycle make it a valuable tool for both research and clinical purposes, with the potential to improve patient care and treatment outcomes.

### 2.3. A Potential Biomarker, CD105

Endoglin, also known as CD105, is a transmembrane glycoprotein primarily found on activated vascular endothelial cells. It plays a crucial role as an accessory protein in the TGF-β receptor system. Structurally, endoglin consists of two subunits, each with a weight of 95 kDa, connected by disulfide bonds. These subunits come together to form a mature 180 kDa protein, and their encoding genes are located on chromosome 9q34 [47,48]. The impact of endoglin on angiogenesis is mediated through its modulation of TGF-β1 signaling. It accomplishes this by binding to its specific receptor and activating the Smad pathways. CD105, in collaboration with TGF-β, has been recognized as a key player in cell proliferation, particularly in association with hypoxia. It is predominantly expressed in activated endothelial cells participating in neoangiogenesis [49]. Immunohistochemical studies consistently reveal strong CD105 expression in the blood vessels within tumor tissues. OSF is characterized by excessive collagen production, resulting in a reduced number of blood vessels and subsequent hypoxia. Thus, evaluating microvessel density (MVD) in OSF could provide valuable insights into its progression towards malignancy [50]. MVD serves as a quantitative measure for assessing tumor angiogenesis and has been linked to tumor aggressiveness and poor prognosis [51]. The assessment of MVD is typically performed in localized regions with a high concentration of blood vessels, referred to as “hot spots.” CD105-positive vascular endothelial cells are identified through brown cytoplasmic staining. MVD analysis is carried out under a microscope at 100× magnification, utilizing specialized methods for examination and interpretation [52,53]. However, it is important to note a limitation in the research: the lack of follow-up on cases that tested positive for CD105 to determine their actual progression into malignancy. Consequently, further studies are essential to investigate CD105′s role in OSF, including follow-up research to assess the malignant potential of the lesion. Additionally, future studies should delve into the molecular level to explore the relationship between TGF-β and CD105 in stimulating angiogenesis in OSF cases. This would help validate the role of angiogenesis in the malignant transformation of OSF [50]. In conclusion, CD105 is a widely used marker for evaluating tumor angiogenesis. Its presence on activated endothelial cells involved in new blood vessel formation makes it a valuable tool for assessing the prognosis and malignant characteristics of various cancers [33,50,52]. Further research in the field is crucial to fully understand its implications in diseases like OSF.

### 2.4. A Potential Biomarker, p63

p63 belongs to the p53 family, sharing significant homology with p53 but playing more intricate physiological roles. These roles encompass the regulation of basal cell proliferation, differentiation, maintenance, and maturation, along with the maturation of stem cell compartments in stratified epithelia [54,55]. There exist more than six protein isoforms, which can be categorized into two groups: isoforms that possess the transcription activation domain (TA isoforms) and isoforms that lack it (Delta N isoforms). Similar to p53, the TA isoforms can activate the transcription of specific target genes, leading to cell cycle arrest and apoptosis induction. The Delta N isoforms exert a dominant negative effect, inhibiting both p53 and TA isoforms from activating transcription, thereby contributing to oncogenic functions that can lead to human cancers [56]. The expression of p63 is exclusively found in cells undergoing the regeneration phase, as it serves as a transcriptional regulator and aids in maintaining the proliferative potential of keratinocytes. It is primarily expressed in the basal layer of stratified oral epithelia under normal circumstances. Consequently, alterations in p63′s immunoexpression may indicate both tumor development and issues with epithelial integrity [57]. Most of the p63 gene expression takes place within the nucleus. These characteristics, linked to changes in epithelial architecture and molecular processes during different stages of precancerous development and its progression into malignancy, can be better identified with the assistance of graph-theoretic features that describe the spatial arrangement of p63+ nuclei with specific histopathological associations. Average roundness factor (ARF), mean tessellation area (MTA), mean tessellation perimeter (MTP), and tessellation disorder of area (TDA) serve to illustrate the diverse conditions of the nucleus in both normal and diseased samples [58]. Quantifying the p63 intensity across the entire oral epithelium using mean grey values for red is feasible. All the basal and suprabasal layers of normal (NORM), oral submucous fibrosis without dysplasia (OSFWOD), and oral submucous fibrosis with dysplasia (OSFWD) tissues were found to express p63. In comparison to normal tissue, there was a significant difference in the level of OSF expression in the entire epithelial p63. In tissue samples from OSFWOD and OSFWD, overexpression was observed. Nevertheless, random basal cells should exhibit a substantial difference in p63 expression between the two disease conditions, as OSFWOD had significantly lower levels of p63 than OSFWD [57]. Although tumors with high p63 expression showed a tendency toward poor recurrence-free and overall survival (*p* = 0.075 and *p* = 0.205, respectively), high p63 expression was not statistically correlated with survival. However, when patients were stratified based on the combined expression of activin A and p63, the group with overexpression of both proteins exhibited the lowest recurrence-free (*p* = 0.005) and overall survival rates [59]. Overexpression of p63 was not only observed in oral cavity cancer (OFC) sample tissues but also found in other cancerous mucosa, such as muscle-invasive bladder cancer, colorectal cancer, and lung disease samples [60].

### 2.5. A Potential Biomarker: miRNA-21

MicroRNAs (miRNAs) constitute a class of non-coding RNA molecules that play a pivotal role in regulating gene expression through various biological processes. Dysregulation of these miRNAs can bestow tumor-promoting abilities upon cells [61]. MiRNAs have emerged as crucial biomarkers for cancer diagnosis and the detection of precancerous lesions, with MiRNA-21 being a prime example [37,62]. MiRNA-21 exerts its influence by modulating the expression of numerous proteins associated with cell survival, apoptosis, and invasion [63]. It acts as a guardian against apoptosis and shields cells from the lethal effects of chemotherapy, thereby facilitating unchecked cellular proliferation [37,63]. The upregulation of MiRNA-21 has been observed in various cancer types, including prostate, breast, liver, brain cancers, and oral squamous cell carcinoma (OSCC) [37]. The consistent overexpression of MiRNA-21 in both cancerous and precancerous conditions has underscored its significance in the development of cancer detection methodologies [64,65]. Salivary levels of MiRNA-21 exhibit notable variation across different stages of oral potentially malignant disorders (OPMD) and oral cancer. This variability positions it as a promising adjunctive diagnostic biomarker, capable of monitoring the various stages of early malignant changes in the oral cavity. In distinct types of OPMD, MiRNA-21 experiences significant upregulation in oral lichen planus and oral leukoplakia, while showing minimal upregulation in OSF [37,62]. In terms of different stages in oral cancer progression, serum MiRNA-21 levels are higher in OSCC patients compared to OSF patients, and the expression of MiRNA-21 significantly increases from stage I to IV in the clinical stages of OSCC [35]. In conclusion, miRNAs play a crucial role in regulating gene expression and promoting cancer development [66,67]. MiRNA-21, in particular, stands out due to its involvement in modulating key biological processes such as cell survival, apoptosis, and invasion [66]. It has been found to be upregulated in various cancers. Its varying levels in saliva across different stages of OPMD and oral cancer make it a promising adjunctive diagnostic biomarker, capable of monitoring the different stages of early malignant changes in the oral cavity. While further research is needed to confirm its practical application, the potential of MiRNA-21 opens new possibilities for cancer diagnosis and monitoring, providing patients with earlier treatment opportunities.

## 3. Discussion

The quest for identifying potential biomarkers in the context of OSF is an endeavor of great significance. This exploration not only holds promise for early disease detection but also provides a window into the intricate mechanisms underpinning the pathogenesis of OSF. In this comprehensive review, we have undertaken a thorough examination of several candidate biomarkers, and in doing so, we have uncovered a wealth of information shedding light on the multifaceted nature of OSF and its progression.

Biomarkers are like keys that can unlock crucial information about a disease [3]. In the case of OSF, these biomarkers serve as molecular signposts that guide us towards a deeper understanding of the condition [3,4]. They have the potential to revolutionize the way we diagnose and manage OSF. By identifying specific biomarkers associated with OSF, we pave the way for early detection, which is often the linchpin in effectively treating a disease. Our exploration into these biomarkers has unveiled a diverse array of molecular candidates, each with its unique attributes and implications.

Ki67, a protein intricately tied to the process of cell proliferation, emerges as a highly promising biomarker in the context of OSF [32]. Its relevance extends beyond mere cell division, as it offers valuable insights into the assessment of disease aggressiveness and the potential prediction of OSF’s progression towards malignancy. However, realizing its full potential as a reliable diagnostic and prognostic tool for OSF requires further research and consideration. While Ki67′s role in cell proliferation is well-established, its specific implications in OSF warrant further investigation. Research should delve into the nuances of Ki67 expression patterns within the context of OSF. This includes exploring variations in Ki67 expression at different stages of OSF, within different subtypes, and in response to various treatment modalities. Understanding how Ki67 relates to the disease’s progression can enhance its diagnostic and prognostic utility. To establish Ki67 as a prognostic tool, future research should focus on correlating Ki67 expression levels with clinical outcomes in OSF patients. Longitudinal studies that track patients over time can reveal whether higher Ki67 expression is associated with a more aggressive disease course, increased risk of malignant transformation, or resistance to treatment. Such correlations can provide valuable guidance for clinical decision-making. Exploring the potential of combining Ki67 with other biomarkers associated with OSF can enhance diagnostic accuracy. Combining multiple biomarkers may provide a more comprehensive assessment of disease severity and progression. For example, integrating Ki67 with CD105 (endoglin), another promising biomarker for OSF, might offer a more robust diagnostic and prognostic toolset.

CD105, also known as endoglin, stands as a compelling biomarker in the realm of OSF due to its pivotal role in angiogenesis, the formation of new blood vessels [33]. The presence of CD105 on activated endothelial cells within OSF lesions provides crucial insights into the vascular changes occurring in this complex condition [33,48]. However, while CD105 holds great promise, further investigations are essential to uncover its full potential and understand its implications in OSF progression towards malignancy. To harness CD105′s diagnostic and prognostic potential fully, future research should delve into its correlations with OSF progression. This involves assessing whether CD105 expression levels change as the disease advances from benign fibrosis to potentially malignant stages. Understanding these correlations can help in risk assessment and clinical decision-making, providing insights into the likelihood of disease escalation. CD105′s role in angiogenesis within the context of OSF merits in-depth exploration. Angiogenesis is a complex process involving the growth of new blood vessels from existing ones, and it is integral to the development and progression of various diseases, including cancer. In OSF, understanding the intricate molecular mechanisms that govern angiogenesis is vital. Research should aim to uncover how CD105 contributes to the formation of new blood vessels within OSF lesions, and whether it promotes disease progression. Furthermore, investigating CD105′s involvement in OSF angiogenesis may open doors to potential therapeutic interventions. If CD105 is found to play a significant role in promoting vascular changes associated with OSF progression towards malignancy, targeting this biomarker or its downstream pathways could represent a therapeutic strategy. This could potentially help mitigate the risk of OSF transitioning to oral cancer.

The p63 protein, a member of the p53 family, is a critical player in the intricate regulation of cell proliferation and differentiation within the oral epithelia [34]. While p63 shows significant promise as a diagnostic biomarker for OSF, further research endeavors are essential to fully unlock its potential. OSF is a multifaceted condition with various clinical presentations and stages. Research should aim to decipher how p63 is expressed differently across these stages and manifestations. For instance, does p63 exhibit higher expression in advanced OSF lesions, in areas of severe fibrosis, or in regions undergoing active pathological changes? Understanding these expression nuances can significantly enhance p63′s role as a diagnostic tool and potentially indicate the stage and severity of OSF. In addition, the spatial distribution of p63 within oral tissues can provide valuable insights into its function and relevance in OSF. Integrating graph-theoretic features into the analysis can offer a more comprehensive understanding of how p63-positive cells are distributed within OSF lesions. By constructing spatial networks that represent cell interactions and distribution patterns, researchers can visualize the spatial signatures associated with different stages and manifestations of OSF. Are there specific spatial arrangements of p63-expressing cells that correlate with fibrotic bands or other histological changes characteristic of OSF? Incorporating graph-theoretic features can unravel spatial cues that contribute to our understanding of p63′s role in OSF pathogenesis.

MiRNA-21, a microRNA intricately involved in cell survival and apoptosis, emerges as a promising adjunctive diagnostic biomarker for OSF [36]. Its varied expression across various oral potentially malignant disorders and oral cancer stages signifies its potential as a valuable tool for monitoring disease progression. However, the practical application of MiRNA-21 in clinical settings necessitates rigorous research and considerations, potentially revolutionizing early detection and intervention strategies in the context of OSF. To fully harness MiRNA-21′s diagnostic potential, it is essential to delve deeper into its variable expression patterns across different stages of OSF and other related oral conditions. Research should aim to elucidate how MiRNA-21 levels change as OSF progresses, and whether specific thresholds can be established to correlate its expression with disease severity. This understanding can provide clinicians with valuable insights into the dynamic nature of OSF and aid in identifying individuals at higher risk of disease progression. To translate MiRNA-21′s promise into practical clinical application, rigorous validation studies are imperative. These studies should involve a diverse and representative cohort of OSF patients, spanning various stages and clinical presentations. The sensitivity, specificity, and predictive value of MiRNA-21 as a diagnostic biomarker should be thoroughly assessed. Additionally, its utility in differentiating OSF from other oral conditions should be explored to ensure its specificity. MiRNA-21 has the potential to revolutionize the early detection and intervention strategies for OSF. By serving as a reliable biomarker, it can facilitate the identification of individuals at higher risk of OSF progression towards malignancy. This early detection can enable timely interventions and personalized treatment approaches, potentially mitigating the impact of OSF on oral and systemic health.

The research into these biomarkers is not merely an academic exercise but a practical pathway to enhance patient care. Early diagnosis, made possible by the identification of reliable biomarkers, can be transformative. It can enable healthcare professionals to intervene at a stage when the disease is more responsive to treatment, potentially improving patient outcomes. Furthermore, understanding how these biomarkers change and evolve as OSF progresses towards malignancy is akin to deciphering a molecular narrative. It guides clinical decision-making, aids in risk assessment, and can even inform tailored therapeutic approaches. Armed with this knowledge, healthcare providers can take a more proactive stance in managing OSF.

To propel the field of OSF biomarker research forward, several critical avenues should be explored. First and foremost, large-scale longitudinal studies are imperative to assess the long-term outcomes of OSF patients who test positive for these biomarkers. Such investigations will provide invaluable insights into the predictive power of these biomarkers concerning malignant transformation. Furthermore, comprehensive mechanistic studies are essential to elucidate the molecular mechanisms underpinning the biomarkers’ roles in OSF pathogenesis. Additionally, the development of non-invasive techniques for biomarker detection, such as saliva-based assays for MiRNA-21, can significantly enhance their clinical utility and patient compliance. Standardization of biomarker assessment protocols and the establishment of diagnostic thresholds are pivotal steps to enable widespread clinical adoption.

## 4. Conclusions

In conclusion, the identification and validation of biomarkers for OSF represent a promising frontier for enhancing early diagnosis and treatment outcomes. The biomarkers highlighted in this review offer valuable insights into OSF’s pathogenesis and progression. Future research should prioritize further validation, mechanistic exploration, and the development of practical diagnostic tools to elevate patient care and mitigate the impact of OSF on both oral and systemic health. Ultimately, a multidisciplinary approach that seamlessly integrates these biomarkers into clinical practice holds immense potential to revolutionize the management of OSF, ultimately improving patient outcomes and quality of life.

## Figures and Tables

**Figure 1 cancers-15-04812-f001:**
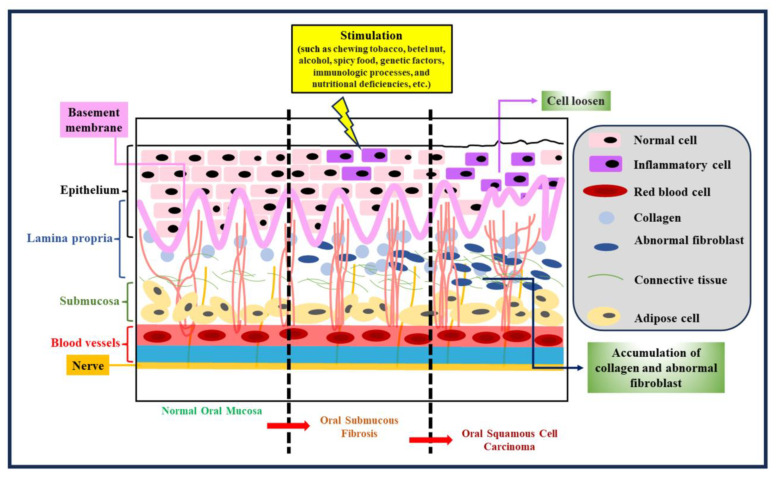
Tissue morphology at different stages. Tissue morphology is pivotal in comprehending the evolution of OSF and its potential transition to oral cancer. OSF advances through distinct stages, each marked by noticeable changes in tissue appearance and structure. Early OSF stages reveal subtle shifts in tissue thickness and texture. Microscopically, altered collagen distribution hints at ongoing collagen deposition. As OSF progresses, the oral mucosa thickens, leading to palpable fibrous bands that limit mouth movement. These bands stem from abnormal collagen accumulation. In advanced OSF stages, oral rigidity causes severe trismus. Tissues visibly toughen, fibrous bands become rigid, and collagen dominates. Timely recognition aids intervention and underscores the need to address OSF’s cause, like quitting betel nut consumption, to impede collagen deposition and thwart cancer progression.

**Table 1 cancers-15-04812-t001:** Biomarkers of OSF specimens.

Specimens		Biomarker in Tissues	Sample Size/Reference
Cells	Cytology	Up: micronuclei in exfoliated buccal cells	leukoplakia (*n* = 15), OSMF (*n* = 15), and OSCC (*n* = 15) [18]
Tissues	DNA	Up: hyper-methylated loci reported in three or more studies included p16, p14, MGMT and DAPK	cell-cycle-control (*n* = 15), DNA-repair (*n* = 7), cell-cycle-signaling (*n* = 4), and apoptosis (*n* = 3) [19]
		Up: Wnt inhibitory factor-1 promoter methylation	OSCC (*n* = 55), OSF (*n* = 45), and normal oral mucosa (*n* = 15) [20]
		Up: secreted frizzled-related proteins (SFRP-1) and SFRP-5	OSCC (*n* = 55), OSF (*n* = 45), and normal oral mucosa (*n* = 15) [21]
		Up: matrix metalloproteinases-3 (MMP-3) polymorphism	OSF (*n* = 5), OSCC (*n* = 5), and normal individuals with tobacco and areca nut habit (*n* = 5) and without (*n* = 5) [22]
	mRNA	Up: Dickkopf WNT signaling pathway inhibitor 3 (DKK3)	OSCC (*n* = 55), OSF (*n* = 45), and normal oral mucosa (*n* = 15) [23]
		Up: profibrotic lncRNA H19 (alert binding of miR-29b and COL1A1	BMF (buccal mucosal fibroblast) cell normal (*n* = 2) and OSF (*n* = 2) [24]
		Up: transforming growth factor β receptor (TGF-βR1 and TGF-βR2)	OSMF (*n* = 33) and normal (*n* = 10) [25]
		Up: LncRNA LINC00974	BMFs and fBMFs (fibrotic buccal mucosa fibroblasts) were retrieved from OSF tissues or the normal counterparts of patients [26]
		Up: miR-1246	OSF tissues (*n* = 20) and BMFs derived from OSF specimen [27]
		Up: miR10-b	OSF specimens (*n* = 20) [28]
		Down: miR-200b	biopsy specimens were taken from the histologically normal oral mucosa and fibrotic mucosa at the time of surgical third molar extraction [29]
		Down: miR-200c	normal (*n* = 25) and OSF (*n* = 25) [30]
	Protein	Up: Ki67	OSMF (*n* = 35), OSCC (*n* = 10), and normal (*n* = 10) [31]
		Up: CD105	paraffin-embedded tissues from normal (*n* = 30) and OSMF (*n* = 50) [32]
		Up: p63	tissue sections of OSCC (*n* = 20), leukoplakia (*n* = 20), OSF (*n* = 20), and normal (*n* = 10) [33]
Serum	Cytology	Up: sister chromatid exchange in lymphocytes	male patients who had the habit of chewing pan for 5 or more years (*n* = 10), male OSF patients who had Pan Parag chewing habit (*n* = 10) and controls without any chewing habit (*n* = 10) [34]
	RNA	Up: miRNA-21	OSCC (*n* = 20), OSF (*n* = 20), and normal (*n* = 40) [35]
	Protein	Up: lactate dehydrogenase (LDH)	OSMF (*n* = 30) and normal (*n* = 30) [36]
		Down: serum protein, globulin	250 participants equally divided in 5 groups (OSMF, OL, NS, OM, and HC) [35]
Saliva	RNA	Up: miRNA-21, miRNA-31	OPMD (*n* = 36) and normal (*n* = 36) [37]
	Protein	Up: LDH	OSMF (*n* = 30) and normal (*n* = 30) [36]
		Down: GPx and SOD	OSF (*n* = 63) and normal (*n* = 63) [37]
	Others	Up: 8-hydroxy-2-deoxyguanosine (8-OHdG) and MDA	OSCC (*n* = 40), OLP lesions (*n* = 40), OL (*n* = 40), OSF (*n* = 40), and control (*n* = 40) [38]

## Data Availability

The data presented in this study are available in this article.

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
