# Peer review of "Exploring Possible Diagnostic Precancerous Biomarkers for Oral Submucous Fibrosis: A Narrative Review"

_cancers, 2023, doi:10.3390/cancers15194812_

Round 1

Reviewer 1 Report

I have reviewed the manuscript “Exploring Possible Diagnostic Precancerous Biomarkers for Oral Submucous Fibrosis” submitted to “Cancers” for publication. In this study, the authors have reviewed the correlation between various biomarkers and OSF; categorized based on cytology and sampling methods. This manuscript fits well within the scope of the journal; it needs some improvements; there are a few suggestions that authors may consider to improve it further:

The use of English language is reasonable, however, there are a number of punctuation and grammatical errors; that should be corrected and rephrased using academic English for a better flow of text for the reader.

Opening paragraph of the intro should not be directly started with oral mucosa.

Authors should clearly mention the search strategy: for instance which articles were included, what were the keywords while searching data bases, which data bases were searchers, did the authors limit the years?

The discussion should be expanded

In line 96, the authors have mentioned about the quality of life, however supporting references are missing. Further references should be included to address the knowledge gap and enhance the effectivity of this review. Evaluating the Oral-Health-Related Quality of Life of Oral Submucous Fibrosis Patients before and after Treatment Using the OHIP-14 Tool. Int J Environ Res Public Health. 2022 Feb 5;19(3):1821. Assessing the Quality of Life of Oral Submucous Fibrosis Patients: A Cross-Sectional Study Using the WHOQOL-BREF Tool. International Journal of Environmental Research and Public Health. 2021; 18(18):9498. A Clinico-Demographic Evaluation of Patients with Oral Submucous Fibrosis: a Cross Sectional Study”, Journal of Pharmaceutical Research International, 33(13), pp. 22–29.

Please confirm about figure 1, is this original and nor review any copywrite permission. 

Minor editing of English language required

Author Response

Manuscript ID: cancers-2627953

Dear Reviewer,

We greatly appreciate your valuable feedback on our manuscript. Your critical comments have been instrumental in identifying key areas that needed further refinement, thereby enhancing the scientific quality of our work. We have diligently addressed each of your comments in the revised manuscript, and a comprehensive list of our responses to your suggestions can be found in the following pages.

We sincerely hope that the revisions we have made will meet the standards for publication in Cancers, and we eagerly anticipate your feedback regarding the suitability of our paper for publication. Your time and expertise are truly appreciated.

Thank you for your kind consideration.

Best regards, 

I-Ta Lee, Ph.D.

School of Dentistry, College of Oral Medicine, Taipei Medical University, Taipei, Taiwan

250 Wuxing St. Taipei 11031, Taiwan

Tel: +886-2-27361661 ext. 5162

Fax: +886-2-27362295

E-mail addresses: [email protected]

Reviewer #1:

  1. The use of English language is reasonable, however, there are a number of punctuation and grammatical errors; that should be corrected and rephrased using academic English for a better flow of text for the reader.

Response:

      We greatly appreciate the reviewer's valuable suggestion. We have meticulously reviewed this manuscript and have made corrections to ensure proper English grammar and readability.

  1. Opening paragraph of the intro should not be directly started with oral mucosa.

Response:

      Thank you very much for the reviewer's suggestion. We have revised this section of the "Introduction".

  1. Authors should clearly mention the search strategy: for instance which articles were included, what were the keywords while searching data bases, which data bases were searchers, did the authors limit the years?

Response:

      Thank you very much for the reviewer’s suggestion. This is a descriptive review aimed at providing comprehensive information on biomarkers associated with oral submucosal fibrosis (OSF). Understanding the pathophysiological processes and potential risk factors for this pre-cancerous condition is crucial. We chose the descriptive review approach because our goal was to summarize the latest research on OSF biomarkers published in the past five years, rather than conducting a systematic quantitative analysis of the existing literature. In the article selection process, we followed these steps:

      Initial Search: We conducted an initial search in the PubMed database using the keywords "fibrosis, oral submucous[MeSH Terms]" and "biomarkers[MeSH Terms]." The choice of these keywords ensured that we covered relevant literature related to OSF and biomarkers.

      Timeframe Restriction: We focused on selecting articles published within the past five years to ensure that we provide the latest research findings. This helps readers gain insights into current research trends and the most recent discoveries.

      Article Screening: We carefully examined the results of the initial search and selected 40 articles that were most relevant to our topic. These articles included case-control studies, which contributed to a deeper understanding of the applications of biomarkers in the diagnosis and prediction of OSF, as well as review articles, which helped us summarize the key findings from past research.

      In summary, our descriptive review aims to provide a comprehensive overview to assist the medical community and researchers in gaining a better understanding of the current status and future research directions concerning OSF and related biomarkers.

  1. The discussion should be expanded.

Response:

      Thank you for the reviewer's suggestion. We have expanded the content in the "Discussion" section.

  1. In line 96, the authors have mentioned about the quality of life, however supporting references are missing. Further references should be included to address the knowledge gap and enhance the effectivity of this review. Evaluating the Oral-Health-Related Quality of Life of Oral Submucous Fibrosis Patients before and after Treatment Using the OHIP-14 Tool. Int J Environ Res Public Health. 2022 Feb 5;19(3):1821. Assessing the Quality of Life of Oral Submucous Fibrosis Patients: A Cross-Sectional Study Using the WHOQOL-BREF Tool. International Journal of Environmental Research and Public Health. 2021; 18(18):9498. A Clinico-Demographic Evaluation of Patients with Oral Submucous Fibrosis: a Cross Sectional Study”, Journal of Pharmaceutical Research International, 33(13), pp. 22-29.

Response:

      Thank you for the reviewer's suggestion. We have added these references in this manuscript.

  1. Please confirm about figure 1, is this original and nor review any copywrite permission.

Response:

      We can confirm that Figure 1 is an original creation. Nevertheless, we have made improvements and visual enhancements to facilitate reader comprehension.

Reviewer 2 Report

This comprehensive review delves into current investigations that explore the correlation between various biomarkers and OSF.

This is a very interesting review article.

Could the authors please specify if this is a descriptive review or if it is a systematic please specify how the selection of the articles was performed?

Was the review registered?

How was the selection strategy?

The limitations of the study were not mentioned.

Please describe why did the authors selected this design for the study and not the usual one?

Moderate

Author Response

Manuscript ID: cancers-2627953

Dear Reviewer,

We greatly appreciate your valuable feedback on our manuscript. Your critical comments have been instrumental in identifying key areas that needed further refinement, thereby enhancing the scientific quality of our work. We have diligently addressed each of your comments in the revised manuscript, and a comprehensive list of our responses to your suggestions can be found in the following pages.

We sincerely hope that the revisions we have made will meet the standards for publication in Cancers, and we eagerly anticipate your feedback regarding the suitability of our paper for publication. Your time and expertise are truly appreciated.

Thank you for your kind consideration.

Best regards, 

I-Ta Lee, Ph.D.

School of Dentistry, College of Oral Medicine, Taipei Medical University, Taipei, Taiwan

250 Wuxing St. Taipei 11031, Taiwan

Tel: +886-2-27361661 ext. 5162

Fax: +886-2-27362295

E-mail addresses: [email protected]

Reviewer #2:

  1. Could the authors please specify if this is a descriptive review or if it is a systematic please specify how the selection of the articles was performed?

Response:

Thank you very much for the reviewer’s suggestion. This is a descriptive review aimed at providing comprehensive information on biomarkers associated with oral submucosal fibrosis (OSF). Understanding the pathophysiological processes and potential risk factors for this pre-cancerous condition is crucial. We chose the descriptive review approach because our goal was to summarize the latest research on OSF biomarkers published in the past five years, rather than conducting a systematic quantitative analysis of the existing literature.

In the article selection process, we followed these steps:

Initial Search: We conducted an initial search in the PubMed database using the keywords "fibrosis, oral submucous[MeSH Terms]" and "biomarkers[MeSH Terms]." The choice of these keywords ensured that we covered relevant literature related to OSF and biomarkers.

Timeframe Restriction: We focused on selecting articles published within the past five years to ensure that we provide the latest research findings. This helps readers gain insights into current research trends and the most recent discoveries.

Article Screening: We carefully examined the results of the initial search and selected 40 articles that were most relevant to our topic. These articles included case-control studies, which contributed to a deeper understanding of the applications of biomarkers in the diagnosis and prediction of OSF, as well as review articles, which helped us summarize the key findings from past research.

In summary, our descriptive review aims to provide a comprehensive overview to assist the medical community and researchers in gaining a better understanding of the current status and future research directions concerning OSF and related biomarkers.

  1. Was the review registered?

Response:

Thank you very much for the reviewer’s suggestion. Since this is a descriptive review, we did not register it.

  1. How was the selection strategy?

Response:

Thank you very much for the reviewer’s suggestion. This is a descriptive review aimed at providing comprehensive information on biomarkers associated with OSF. In the article selection process, we followed these steps:

Initial Search: We conducted an initial search in the PubMed database using the keywords "fibrosis, oral submucous[MeSH Terms]" and "biomarkers[MeSH Terms]." The choice of these keywords ensured that we covered relevant literature related to OSF and biomarkers.

Timeframe Restriction: We focused on selecting articles published within the past five years to ensure that we provide the latest research findings. This helps readers gain insights into current research trends and the most recent discoveries.

Article Screening: We carefully examined the results of the initial search and selected 40 articles that were most relevant to our topic. These articles included case-control studies, which contributed to a deeper understanding of the applications of biomarkers in the diagnosis and prediction of OSF, as well as review articles, which helped us summarize the key findings from past research.

  1. The limitations of the study were not mentioned.

Response:

Thank you very much for the reviewer’s suggestion. This is a descriptive review aimed at providing comprehensive information on biomarkers associated with OSF. In the article selection process, we followed these steps:

Initial Search: We conducted an initial search in the PubMed database using the keywords "fibrosis, oral submucous[MeSH Terms]" and "biomarkers[MeSH Terms]." The choice of these keywords ensured that we covered relevant literature related to OSF and biomarkers.

Timeframe Restriction: We focused on selecting articles published within the past five years to ensure that we provide the latest research findings. This helps readers gain insights into current research trends and the most recent discoveries.

Article Screening: We carefully examined the results of the initial search and selected 40 articles that were most relevant to our topic. These articles included case-control studies, which contributed to a deeper understanding of the applications of biomarkers in the diagnosis and prediction of OSF, as well as review articles, which helped us summarize the key findings from past research.

  1. Please describe why did the authors selected this design for the study and not the usual one?

Response:

Thank you very much for the reviewer’s suggestion. This is a descriptive review aimed at providing comprehensive information on biomarkers associated with oral submucosal fibrosis (OSF). Understanding the pathophysiological processes and potential risk factors for this pre-cancerous condition is crucial. We chose the descriptive review approach because our goal was to summarize the latest research on OSF biomarkers published in the past five years, rather than conducting a systematic quantitative analysis of the existing literature. In summary, our descriptive review aims to provide a comprehensive overview to assist the medical community and researchers in gaining a better understanding of the current status and future research directions concerning OSF and related biomarkers.

Round 2

Reviewer 1 Report

Thank you for revising the manuscript.

Minor errors